# Effects of Different Trunk Training Methods for Chronic Low Back Pain: A Meta-Analysis

**DOI:** 10.3390/ijerph19052863

**Published:** 2022-03-01

**Authors:** Dhananjaya Sutanto, Robin S. T. Ho, Eric T. C. Poon, Yijian Yang, Stephen H. S. Wong

**Affiliations:** 1Department of Sports Science and Physical Education, The Chinese University of Hong Kong, Shatin, Hong Kong; damien.sutanto@link.cuhk.edu.hk (D.S.); robinho@cuhk.edu.hk (R.S.T.H.); yyang@cuhk.edu.hk (Y.Y.); 2Department of Health and Physical Education, The Education University of Hong Kong, Taipo, Hong Kong; ericpoon@eduhk.hk

**Keywords:** low back pain, rehabilitation, exercise therapy

## Abstract

We conducted a systematic review and meta-analysis comparing motor control, isometric, and isotonic trunk training intervention for pain, disability, and re-injury risk reduction in chronic low back pain patients. The EMBASE, MEDLINE, CENTRAL, PsycINFO, SPORTDiscus, and CINAHL databases were searched from inception until 25 February 2021 for chronic low back pain intervention based on any trunk training. Outcomes include the Oswestry Disability Index (ODI) and Roland Morris Disability Questionnaire (RMDQ) for disability, the Numerical Pain Rating Scale (NPRS) for pain, and the Sorensen Test (ST) for future risk of re-injury. Isometric training was superior to the control with a mean difference (MD) = −1.66, 95% confidence interval (CI) [−2.30, −1.01] in pain reduction; MD = −7.94, 95% CI [−10.29, −5.59] in ODI; MD = −3.21, 95% CI [−4.83, −1.60] in RMDQ; and MD = 56.35 s, 95% CI [51.81 s, 60.90 s] in ST. Motor control was superior to the control with a MD = −2.44, 95% CI [−3.10, −1.79] in NPRS; MD = −8.32, 95% CI [−13.43, −3.22] in ODI; and MD = −3.58, 95% CI [−5.13, −2.03] in RMDQ. Isometric and motor control methods can effectively reduce pain and disability, with the isometric method reducing re-injury risk.

## 1. Introduction

Most people experience low back pain within their lifetime [1,2], with mild to severe symptoms [3]. Physicians can identify the cause of pain 10% of the time, often giving a non-specific diagnosis [1]. Up to one-third of non-specific low back pain patients develop chronic symptoms persisting over three months [4], with a global prevalence of around 25% of the working-age population [4,5]. Trunk muscle training is often prescribed for the treatment of chronic low back pain (CLBP). Trunk muscle tension increases the spine’s ability to remain in a neutral pose under load [6] whereas spine posture and movement away from a neutral pose under load can increase the risk of spine tissue damage [7,8,9]. There is considerable trunk muscle training with no explicit clinical guidelines on the types considered effective. With CLBP significantly increasing the risk of co-morbidities including musculoskeletal, neuropathic, and psychological issues [10], we need to better understand the effectiveness of different trunk muscle training methods.

Trunk training comes in many forms [11,12,13], and most can be classified based on the biomechanical properties and training focus as:Isometric (IM) training: loading the spine while the trunk muscles contract to maintain the spine in a neutral position [14]. Plank, bird-dog, and side bridge are some isometric trunk training examples.Isotonic (IT) training: moving the lumbar spine through a range of motion while under load, both eccentrically and concentrically [13,15]. Sit-up and back extension focused on segmental spine movement are examples of isotonic trunk training.Motor control (MC) training: isolated activation of the deep trunk musculature targeting the transverse abdominis, lumbar multifidus, diaphragm, and pelvic floor [12,16]. Some examples of this include focusing on abdominal drawing-in manoeuvre or abdominal hollowing in isolation at different positions.

Similar exercises with different training methods result in different muscle activation [17]. Altered trunk muscle activation patterns may cause CLBP, hence the MC method prioritises isolated deep activation [12]. Some types of CLBP injury can be exacerbated by a full range of motion [9] or movement under compression [8,18], hence the IM method focuses on training trunk muscle endurance while minimising spine loading [19].

Existing primary studies comparing different types of trunk training have resulted in inconsistent findings. MC could be more [20] or less effective [21] than IM in pain reduction based on the Numerical Pain Rating Scale (NPRS); more [22] or less effective [23] than IT in pain reduction; or just as [24] or more [25] effective than the control group in disability reduction based on the Roland Morris Disability Questionnaire (RMDQ). In some cases, combining different training methods also yielded inconclusive results [26,27]. Recent meta-analysis either did not specify details on the measured outcome used for disability [28], or combined the RMDQ and Oswestry Disability Index (ODI) measurements into a single outcome [27]. RMDQ and ODI have different sensitivity depending on the patient’s disability level [29], and combining them may not be appropriate. There is a lack of evidence on the comparative effectiveness of different trunk training methods in pain and disability reduction, along with the future risk of CLBP re-injury.

This study aimed to evaluate and synthesise the comparative effectiveness of MC, IM, and IT trunk training methods using a meta-analysis based on validated outcomes. This novel approach would provide clinical practitioners with more specific guidelines on CLBP training prescription. Subgroup analysis was performed to compare the effect of training duration and patient age on different trunk training methods. The subgroup analysis can provide more insights on how training duration and patient age can affect trunk training effectiveness.

## 2. Materials and Methods

### 2.1. Experimental Approach to the Problem

This study was in accordance with the Preferred Reporting Items for Systematic Reviews and Meta-Analyses (PRISMA) guidelines on systematic reviews and meta-analyses [30]. The protocol was registered on the International Prospective Register of Systematic Reviews (PROSPERO; Registration No. CRD42020168972).

Electronic databases (EMBASE, MEDLINE, PsycINFO, CENTRAL, CINAHL, and SPORTDiscus) were searched from inception until 25 February 2021, and peer-reviewed. No language restriction was applied, with a complete search strategy based on a past Cochrane review [31] available in Appendix A. Non-English journal articles were translated into English using Google Translate before data extraction. Data extracted from each study included subject demographics, inclusion and exclusion criteria, intervention details, and outcome measures.

### 2.2. Subjects

This study focused on the working-age population (19–60 years) as a significant percentage of this population experience CLBP [5,32]. The 19–25 year old population group may have different causes of CLBP and respond differently to training treatment compared to the 50–60 year old population group. Due to the non-specific nature of the CLBP diagnosis given to the 19–60 year population group, recent meta-analyses on CLBP intervention [26,27,28] have focused on this population as a whole. Many intervention studies have either used a similar age range [21,23] or do not specify any age range [33,34] in their subject recruitment criteria. CLBP is defined as persistent low back pain for at least 12 weeks [3,35] or as diagnosed by a clinician. Studies on patients with osteoarthritis, cancer, or cardiovascular disease such as claudication were excluded as their underlying condition may cause CLBP [36,37,38]. Randomised controlled trials (RCT) with patients that had undergone major spinal surgery were excluded as surgery may change the muscle, fascia, or neural structure around the lumbar area [39,40].

### 2.3. Procedures

Interventions compared for this study were trunk training that could be classified as IM, IT, or MC. Training that could not be exclusively classified as either was excluded [41,42]. Interventions combining trunk training with aerobic or limb strength training exceeding 15 min were excluded as aerobics and limb strength training may reduce the pain and disability of CLBP patients [28,43].

The included RCTs had to contain either a control or a different trunk training group as a comparator in the pre- and post-intervention. The control included passive interventions considered ineffective for CLBP, placebo intervention, or simple advice such as maintaining active daily living or exercise avoidance. Passive interventions included transcutaneous electrical nerve stimulation (TENS) [44], ultrasound [45], and patient education only [46]. Placebo intervention included detuned ultrasound, TENS, and sham massage. Home exercise prescriptions were not considered as a valid control because patient training adherence may differ and cause high measurement variability. Flexibility and mobility are not significant predictors of future CLBP [47] and have no significant effect on pain and disability reduction [28], hence brief warm-up stretching and mobility exercises on the intervention or control were acceptable.

Outcomes can be classified as Patient-Reported Outcome Measures (PROM) or Patient Performance Test (PPT) based on the COSMIN guidelines for CLBP. A recent systematic review concluded that ST has high test–retest, intra-rater, and inter-rater reliability for PPT [48]. ST is inversely correlated to CLBP risk across the study population of interest [49,50] and was chosen as the PPT for the meta-analysis.

PROMs are outcomes based on the participant’s subjective responses. A recent Delphi study concluded that NPRS, ODI, and RMDQ are the most widely accepted PROMs for CLBP intervention [51]. This review only included clinical intervention studies excluding cohort studies, case studies, commentaries, and editorials. Patients scoring 61% or above on the ODI were considered to have a crippling disability [52] and should be receiving positive intervention instead of physical training [29]. Hence, studies with patients from this group were excluded from the analysis.

Current international guidelines for the treatment of CLBP do not recommend CLBP patient subgrouping [3,35,53,54]. RCTs that use specific classification to separate patients into different treatments were excluded to ensure the external validity of the meta-analysis. Article titles and abstracts identified from the search results were independently assessed by two reviewers, with the primary research data exported to Endnote X9.2 build 13,018. Relevant grey literature was searched for related trial data. Full-text articles were screened independently by both reviewers, with a third reviewer adjudicating any disagreement. Published articles with the most relevant outcomes were included in the analysis for multiple publications from a single RCT. Publication data (author, year, and origin), study design (patients and groups numbers), intervention, and outcome from included RCTs were extracted to Table A1 in Appendix B, with the primary authors being contacted for missing data.

Review Manager v5.4.1 (Cochrane Collaboration, Copenhagen, Denmark) was used for the statistical analyses. Before input in the meta-analysis, pre- and post-intervention mean differences (MD) and standard deviations (SD) from the included studies were converted to change MD and SD based on Cochrane handbook section 16.1.3.2. Significance was set for α at 5% and 95% confidence interval (CI), and all analyses used the random effect model. Pain VAS reported as a score of 0–100 was standardised to NPRS 0–10. Weight column indicated in the meta-analysis result in Review Manager indicates the effect percentage from a particular study towards the overall MD. Green squares on meta-analysis result graphically indicates the effect of each individual study while black squares indicate the overall effect of the combined studies. Heterogeneity as I2 was classified as low (~25%), moderate (~50%), and high (~75%) [55]. High heterogeneity could be due to publication bias, methodological issues, or clinical differences. Methodological issues were investigated under the risk of bias assessment. The clinical difference was investigated using subgroup analysis. Detailed significant and insignificant results are displayed in separate figures.

The Cochrane Risk of Bias 2.0 tool was used to assess the randomisation, assessment, missing outcome, measurement outcome, and reporting outcome bias [56]. Two researchers independently evaluated and resolved differences through discussion. Sensitivity analysis was conducted by removing the high risk of bias studies. Result certainty was assessed based on the result and heterogeneity change after sensitivity analysis.

The trunk training clinical trial durations ranged from one [33] to twelve months [57]. Training duration subgroup analysis could justify a longer duration training prescription for severely affected CLBP patients. Patients under 40 may have a 3.7 times higher chance of pain and disability reduction than older patients [58]. Some intervention studies recruited patients with an age range that overlapped 40 [21,34]. Patient mean age was used to group the included studies. Studies were grouped into under-40, 40–45, and over-45 years old, with the analysis only comparing the under-40 and over 45-groups. Effects of ageing on human physiology and related training adaptation is gradual and non-uniform. Some studies with participants with a mean age of 40–45 may have an overall physiology closer to those under 40, while in other studies, the overall physiology was closer to those over 45. The 40–45 age group was not analysed to remove uncertainty in their classification as more similar to those under-40 or above-45 in the subgroup analysis. Intervention studies with the participants’ mean age of different group belonging to different classification were excluded. This subgroup analysis provided an insight into the effect of patient age on CLBP training effectiveness.

## 3. Results

The literature search identified 10,846 citations with 10,372 citations excluded after screening the title and abstracts and full-text screening of the remaining 476 citations. One study with a related intervention and outcome was excluded as all the subjects later received surgical intervention prior to post-intervention outcome measurement [59]. Studies with non-chronic low back pain subjects were excluded [60,61] as the majority of low back pain cases resolve spontaneously within 6 weeks [62]. Forty-seven RCTs (N = 2299) were included in the meta-analysis. A PRISMA flowchart [30] of the RCT search and selection results is shown in Figure 1.

Individual study characteristics of the included RCTs are mentioned in Table A1 in Appendix B. Included RCTs were from 19 countries: 26 studies from Asia, 15 from Europe, five from America, and one from Australia. One study was in the Korean language only, while the remaining 44 were in English and another language or were written only in English. In studies with a control, seven had no details on the control intervention, five received exercise avoidance advice, four received patient education, four received passive treatment, three received a passive placebo treatment, and three were put on a waiting list. The number of patients per intervention group at baseline ranged from 5–84, with 22 of the 47 RCTs having 20–40 patients per group. Six of the included RCTs had over 40 patients per group.

Two studies recruited male patients only. Five studies did not specify their patient gender demographics. Ten studies exclusively recruited female patients and 30 studies recruited both male and female patients. Nine of the studies used four weeks as the duration for training intervention, 15 used six weeks, 14 used eight weeks, two used 10 weeks, and seven used 12 weeks. Sixteen studies used patients with a mean age below 40, nine studies used patients with a mean age 40–45, and 12 studies used patients with a mean age above 45 years old. Seven of the included studies had intervention arms belonging to different age groups and three did not have information on their recruited subject age.

IM intervention was more effective than the control in reducing pain as measured by NPRS (Figure 2, first row), MD = −1.66, 95% CI [−2.30, −1.01], I2 = 90%, *p* < 0.01. IM was superior to the control in disability reduction as measured by ODI (Figure 2, second row), MD = −7.94, 95% CI [−10.29, −5.59], I2 = 60%, *p* < 0.01, and RMDQ (Figure 2, third row), MD = −3.21, 95% CI [−4.83, −1.60], I2 = 84%, *p* < 0.01. IM intervention increased trunk extensor endurance (Figure 2, 17^th^ row) compared to that of the control with MD = 56.35, 95% CI [51.81, 60.90], I2 = 0%, *p* < 0.01.

MC was more effective in reducing pain than the control as measured by the NPRS (Figure 2, first row), MD = −2.44, 95% CI [−3.10, −1.79], I2 = 79%, *p* < 0.01. MC was superior in controlling disability reduction as measured by the ODI (Figure 3, second row), MD = −8.32, 95% CI [−13.43, −3.22], I2 = 43%, *p* < 0.01 and as measured by RMDQ (Figure 3, third row), MD = −3.58, 95% CI [−5.13, −2.03], I2 = 47%, *p* < 0.01.

A pairwise meta-analysis comparing different training methods resulted in MC being superior to IT on the NPRS (Figure 4, first row), MD = −0.84, 95% CI [−1.56 to −0.11], I2 = 86%, *p* = 0.02, and ODI (Figure 4, second row), MD = −4.66, 95% CI [−7.67, −1.65], I2 = 84%, *p* < 0.01. MC was superior to IM in disability reduction based on the ODI (Figure 4, third row), MD = −5.95, 95% CI [−10.77, −1.12], I2 = 88%, *p* = 0.02. The difference was not significant as measured with the NPRS (Figure 4, fourth row), MD = −0.09, 95% CI [−0.42, 0.24], I2 = 67%, *p* = 0.61, and RMDQ (Figure 4, fifth row), MD = 0.78, 95% CI [−0.66, 2.22], I2 = 22%, *p* = 0.29.

IT intervention did not result in significant NPRS reduction compared to the control (Figure 5, first row), with MD = −0.87, 95% CI [−2.05, 0.31], I2 = 74%, *p* = 0.15, while IM and IT intervention were not significantly different in the NPRS (Figure 5, second row) with MD = 0.19, 95% CI [−0.36, 0.74], I2 = 65%, *p* = 0.50. IT significantly reduced disability as measured with ODI compared to the control (Figure 5, third row) with MD = −11.22, 95% CI [−18.01, −4.42], I2 = 77%, *p* = 0.001. In addition, IT was not significantly different to IM in ODI reduction (Figure 5, fourth row) with MD = 0.25, 95% CI [−2.24, 2.74], I2 = 74%, *p* = 0.85. MC to IT comparison in the RMDQ outcome did not show any significant difference (Figure 5, fifth row), with MD = 0.42, 95% CI [−0.83, 1.67], I2 = 0%, *p* = 0.51. The IT to control comparison resulted in the largest disability (ODI) reduction (MD = −11.22) while MC was more effective than IT (MD = −4.66).

### 3.1. Sensitivity Analysis

Figure 6 lists the included studies along with the risk of bias in the five bias domains. Studies with low risk of bias in all five domains were judged as having low overall risk of bias. Studies with concerns in their methodology in two or less domains were judged as having some concerns in the overall risk of bias. Studies with three or more domains having methodological concerns or with high risk of bias in one of the domains were judged as having high risk in overall risk of bias. Overall, four studies had a low risk of bias, 21 had some concerns, and 21 had a high risk of bias. Ninety percent of the concerns in the randomisation bias were due to a lack of allocation concealment in the study report. Over 60% of the studies had some concerns on measurement outcome bias due to the lack of assessor blinding on group allocation, with PROMs being subjective in nature. The 21 studies with a high risk of bias were removed in the sensitivity analysis, while another study [63] was removed from the NPRS results due to missing data.

Difference in MC and IM effect on disability reduction as measured by the ODI became non-significant (Appendix C.1, sixth row), MD = −0.56, 95% CI [−1.86, 0.75], I2 = 0%, *p* = 0.40. The difference remained non-significant as measured with the NPRS (Appendix C.1, first row), MD = 0.08, 95% CI [−0.21, 0.37], I2 = 24%, *p* = 0.58. IM remained more effective than the control based on the NPRS (Appendix C.1, second row), MD = −1.55, 95% CI [−2.28, −0.83], I2 = 92%, *p* < 0.01, and ODI (Appendix C.1, seventh row), MD = −6.43, 95% CI [−7.80, −5.07]; I2 = 0%, *p* < 0.01. MC remained more effective than the control in pain reduction as measured by the NPRS (Appendix C.1, third row), MD = −2.12, 95% CI [−2.89, −1.35], I2 = 83%, *p* < 0.01. In addition, MC remained more effective in disability reduction as measured by the ODI (Appendix C.1, eighth row), MD = −10.46, 95% CI [−15.25, −5.66], I2 = 0%, *p* < 0.01, and RMDQ (Appendix C.1, eleventh row), MD = −3.44, 95% CI [−5.24, −1.63], I2 = 55%, *p* < 0.01. MC was no longer superior to IT in NPRS reduction (Appendix C.1, fourth row) with MD = −0.59, 95% CI [−1.49, 0.32], I2 = 71%, *p* = 0.20, while remaining superior in ODI reduction (Appendix C.1, ninth row), with MD = −2.53, 95% CI [−3.58, −1.49]; I2 = 0%, *p* = 0.01.

Regarding IM to IT comparison in the NPRS outcome, four of the six included studies were removed due to a high risk of bias with a sensitivity analysis showing no significant difference (Appendix C.1, fifth row), MD = 0.76, 95% CI [−1.03, 2.55], I2 = 92%, *p* = 0.40. In the IM to IT comparison in the ODI outcome, three of the six included studies had a high risk of bias according to risk of bias analysis, with no significant difference in sensitivity analysis (Appendix C.1, tenth row), MD = 0.76, 95% CI [−1.03, 2.55], I2 = 92%, *p* = 0.40.

### 3.2. Training Duration Subgroup Analysis

The difference between IM and MC effects on the NPRS (Appendix C.2, first row) outcome was insignificant for under eight weeks of intervention, MD = −0.25, 95% CI [−0.74, 0.24], I2 = 74%, *p* = 0.32, and for eight or more weeks of intervention, MD = 0.14, 95% CI [−0.28, 0.55], I2 = 47%, *p* = 0.52. Compared to a total heterogeneity of 67%, subgroup analysis resulted in lower heterogeneity in long-term intervention and higher heterogeneity in short-term intervention with no significant difference between the two groups, *p* = 0.24.

IM comparison with the control in the NPRS outcome (Appendix C.2, second row) was significant for under eight weeks, MD = −1.10, 95% CI [−1.65, −0.54], I2 = 84%, *p* < 0.01 and for eight or more weeks of training duration, MD = −2.58, 95% CI [−3.32, −1.83], I2 = 54%, *p* < 0.01. Compared to the total heterogeneity of 90%, subgroup analysis resulted in lower heterogeneity in both subgroups and greater pain reduction in longer duration interventions (*p* < 0.01).

MC comparison with the control (Appendix C.2, third row) resulted in significant NPRS reduction for under eight weeks of intervention, MD = −2.51, 95% CI [−4.12, −0.89], I2 = 91%, *p* < 0.01 and for eight or more weeks of intervention, MD = −2.47, 95% CI [−3.15, −1.79], I2 = 64%, *p* < 0.01. Compared to a total heterogeneity of 82%, subgroup analysis resulted in lower heterogeneity in long-term intervention and higher heterogeneity in short-term intervention with no significant difference between the two groups (*p* = 0.86).

IM comparison with the control in ODI outcome (Appendix C.2, fourth row) was significant for a training duration of under eight weeks, MD = −6.17, 95% CI [−7.61, −4.74], I2 = 0%, *p* < 0.01 and for eight or more weeks, MD = −12.07, 95% CI [−18.72, −5.41], I2 = 75%, *p* < 0.01. Compared to a total heterogeneity of 60%, subgroup analysis resulted in lower heterogeneity in shorter duration while increasing heterogeneity in longer duration, with no significant difference between the two groups (*p* = 0.09).

### 3.3. Age Subgroup Analysis

Subgroup analysis on IM intervention indicated that IM was effective in pain (NPRS) reduction in all age groups (Appendix C.3, first row). Patients under 40 years experienced greater pain reduction, MD = −1.99, 95% CI [−2.44, −1.53], I2 = 0%, *p* < 0.01, compared to patients over 45 years of age, MD = −1.32, 95% CI [−2.46, −0.18], I2 = 73%, *p* = 0.02. Heterogeneity of both groups was lower than the total heterogeneity of I2 = 84% with a significant difference within the groups (*p* = 0.01). IM intervention was also effective in disability reduction (ODI) among all age groups (Appendix C.3, third row). Patients under 40 years experienced similar disability reduction, MD = −7.61, 95% CI [−10.88, −4.33], I2 = 0%, *p* < 0.01, compared with that of patients over 45 years of age, MD = −10.16, 95% CI [−18.66, −1.66], I2 = 87%, *p* = 0.02. The heterogeneity of the under 40 group was lower, I2 = 0%, than the total heterogeneity of I2 = 65% with the highest heterogeneity in the over 45 group, I2 = 87%. There was no significant difference within the groups (*p* = 0.49).

Subgroup analysis based on age indicated that MC intervention was effective in pain reduction (NPRS) in all age groups (Appendix C.3, second row). Patients under 40 years experienced significantly greater pain reduction, MD = −3.11, 95% CI [−3.70, −2.52], I2 = 67%, *p* < 0.01, compared to that in patients over 45 years of age, MD = −1.39, 95% CI [−2.40, −0.39], I2 = 68%, *p* < 0.01. Heterogeneity of both groups were lower than the total heterogeneity of I2 = 79%, with a significant difference between the subgroups (*p* = 0.02).

## 4. Discussion

Both IM and MC interventions resulted in clinically significant pain and disability reduction in CLBP patients according to the ACP definition [32]. IM methods may also be effective in CLBP re-injury risk reduction based on increased trunk extensor endurance [47]. All three intervention groups have often been grouped as one in past syntheses [27,28], which resulted in lower pain or disability reduction as IT intervention was not effective in pain (NPRS) and disability (RMDQ) reduction. Sensitivity analysis resulted in IM and MC interventions being effective in pain and disability reduction. IT was ineffective in reducing pain (NPRS), possibly due to the training loading that imitates some patient-specific spine injury mechanisms [7,8,9,18].

Past intervention studies have indicated that single postural re-education intervention can reduce pain and disability in acute and chronic low back pain patient [64,65]. This result is consistent with the recent meta-analysis on postural re-education for CLBP [66]. MC and IM may be equally effective as both focus on developing the muscular endurance to hold the spine in a neutral position including during limb movement progressions that may have a similar effect with global postural re-education. Most included RCTs comparing both interventions equalised training intensity by having an IM group training duration 30–50% less than that of the MC group, which may cause both groups to have similar outcomes [67,68,69,70].

Inconsistent results in pair-wise meta-analyses with the IT method could be due to the small number of included studies within some pairwise meta-analyses, high risk of bias in some of the included studies, no standard in trunk training frequency and duration, and variability in the recruited patient age group and training duration.

Only three of the included studies used ST in comparing MC and IM [21,34] and only two included studies comparing IT to IM [63,71]. Standardisation and use of a select set of objective outcomes would provide better comparison in future meta-analyses.

Subgroup analysis on CLBP patients trained with IM methods indicated that a longer training duration resulted in further pain reduction, while disability reduction was not significantly different. This indicates that IM intervention may reduce disability earlier than pain. The training duration did not significantly affect pain reduction in MC intervention, indicating that other factors such as clinician skill or difference in training prescription may have an impact that is more significant. Age subgroup analysis indicated that both MC and IM intervention was effective in all age groups, with patients under 40 years experiencing greater pain reduction compared to those over 45 years of age. This could be because older patients require a higher training stimulus to achieve comparable muscular adaptation as that in younger patients [72].

Limitations of the current meta-analysis include a lack of analysis on gender difference, effects of training intensity, and comparison between isolated trunk training and progression with limb movement due to insufficient data. The effect of patient grouping based on specific assessment exceeded the scope of this study. Future research could focus on a single training method in one intervention group to enable a better understanding of the effects of a specific trunk training method on CLBP patient outcomes. In addition, future RCTs should consider incorporating the NPRS, ODI, RMDQ, and ST measurement and follow the CONSORT [73] guidelines to reduce the risk of bias and increase methodological transparency. Future meta-analyses should consider the difference in the recruited patient age demographic and training duration when comparing the different types of CLBP interventions. Other outcome measures and multi-modal interventions may be useful, however, these were excluded to limit the scope of this study.

## 5. Conclusions

Clinicians can prescribe trunk muscle training, focusing on deep abdominal muscle activation (MC method) such as the abdominal draw-in-manoeuvre or isometric trunk muscle activation (IM method) such as the plank for patients with CLBP. Both training approaches can be effective as both methods train the trunk muscle endurance to hold the spine in a neutral position including during active daily living. As the spine in a neutral position is more resilient to tissue injury, CLBP patients trained in the MC and IM methods could gradually experience pain and disability reduction. Trunk muscle training focusing on spine movement (IT method) such as sit-ups may be less effective in pain reduction as it does not train CLBP patients to hold their spine in a neutral position.

Short-term IM training intervention from four to six weeks can result in a pain and disability reduction. CLBP patients with a larger pain score can experience a larger pain reduction with a longer IM intervention of at least eight weeks. Both IM and MC methods may result in larger pain reduction in patients under 40 compared to those over 45. Future CLBP intervention studies should use participants with the same mean age on different groups while future meta-analysis should consider limiting the age range of the included studies’ populations. Further study on the effect of ageing on CLBP training adaptation, and how to adapt training prescription according to CLBP patient age can be investigated in future studies.

## Figures and Tables

**Figure 1 ijerph-19-02863-f001:**
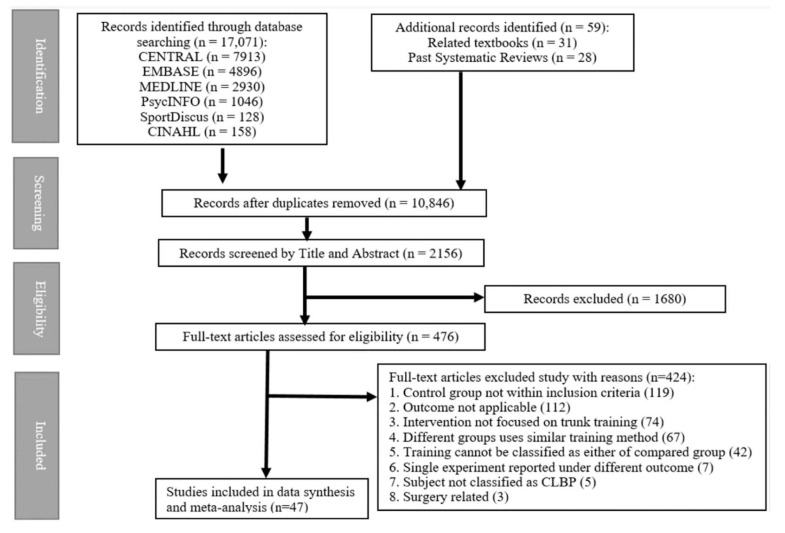
Flowchart of literature selection in the systematic reviews on trunk muscle training for chronic low back pain.

**Figure 2 ijerph-19-02863-f002:**
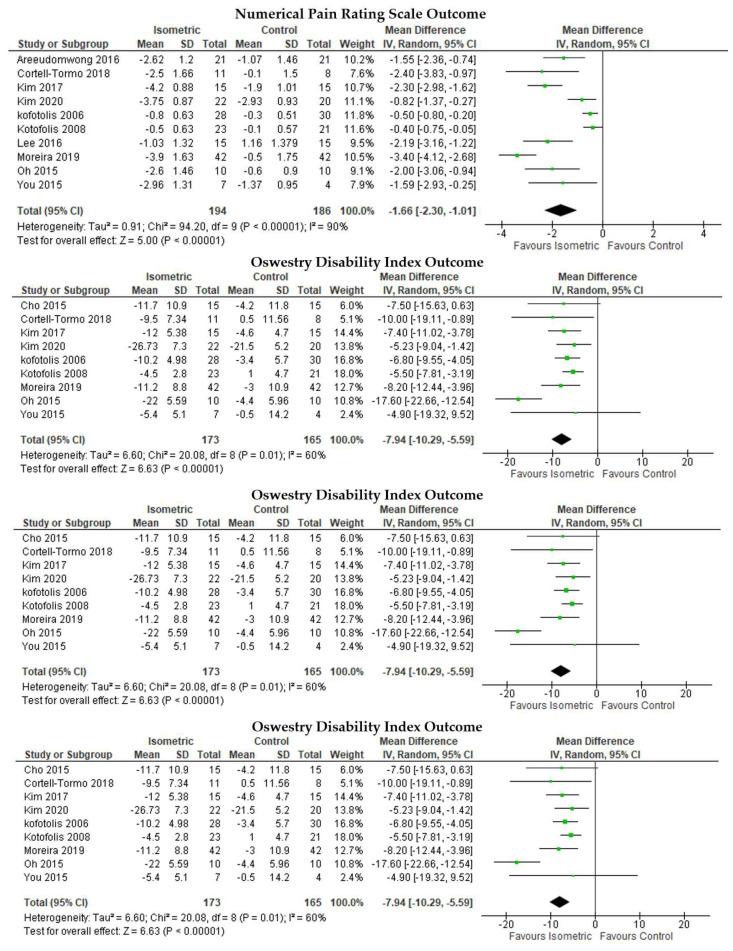
Pairwise meta-analyses on the effectiveness of isometric trunk muscle training compared to the control for chronic low back pain.

**Figure 3 ijerph-19-02863-f003:**
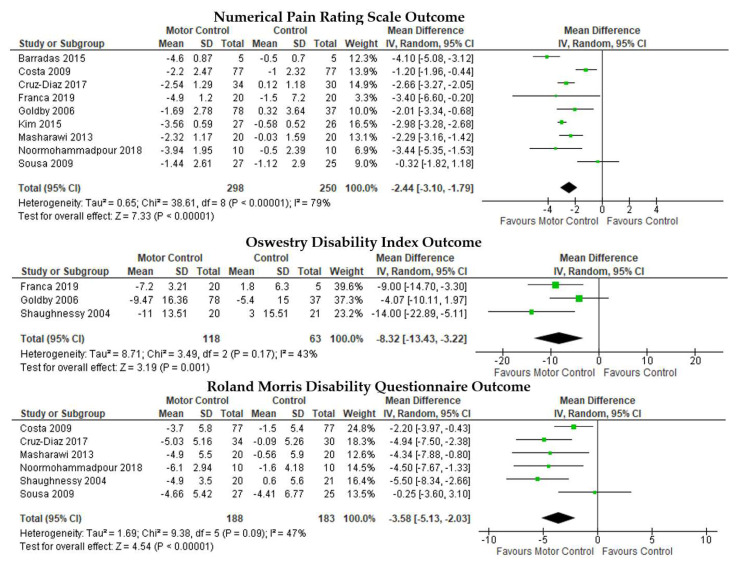
Pairwise meta-analyses on the effectiveness of motor control trunk muscle training compared to the control for chronic low back pain.

**Figure 4 ijerph-19-02863-f004:**
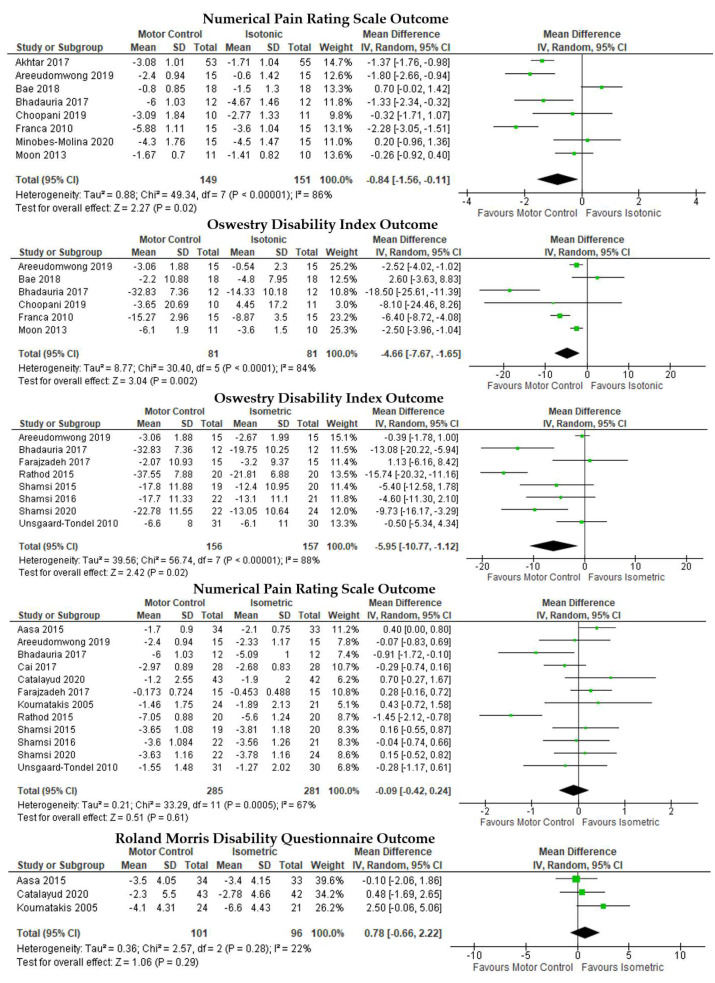
Pairwise meta-analyses on the comparative effectiveness of different trunk muscle training methods for chronic low back pain.

**Figure 5 ijerph-19-02863-f005:**
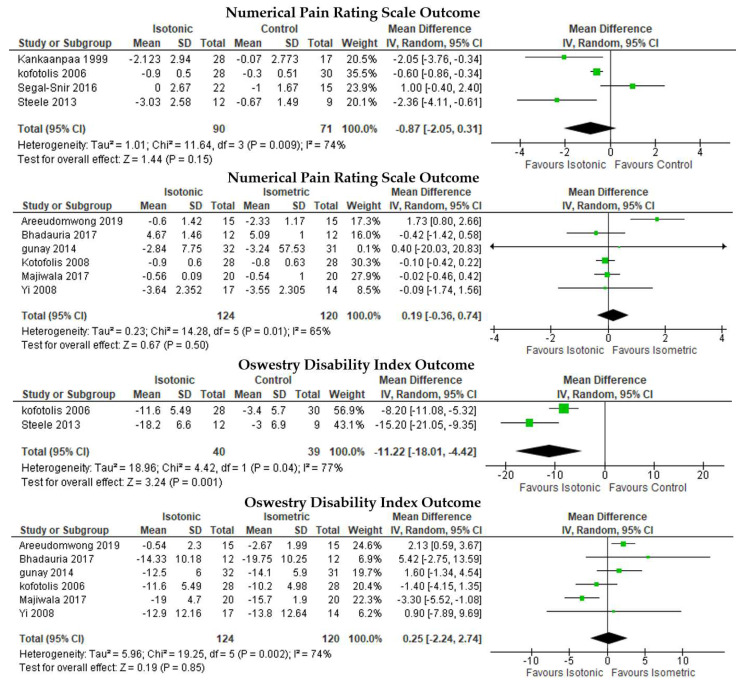
Pairwise meta-analyses with no significant results.

**Figure 6 ijerph-19-02863-f006:**
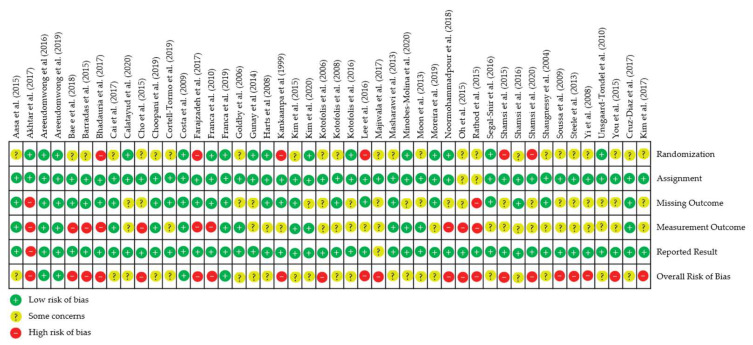
Risk of bias among the included randomised controlled trials.

## Data Availability

The data in this study can be provided upon reasonable request to the corresponding author.

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
