# Peer review of "Effects of Different Trunk Training Methods for Chronic Low Back Pain: A Meta-Analysis"

_ijerph, 2022, doi:10.3390/ijerph19052863_

Round 1
Reviewer 1 Report
Dear authors,
It's an important area to research, the overall is satisfying, however several things should be explained before further steps:
- I suggest to modify the title and move the "meta-analysis" to the end
- citation for "often giving 25 a non-specific diagnosis" is needed,
- why age range is so wide? It is always difficult to compare teenager (19) with somebody at 60, no matter what aspect you consider. Please explain the appropriate statistics for it.
- persistent is an inappropriate definition of CLBP, while chronic might not be persistent, please specify in the text
- all the graphics and tables look not well organized, a bit messy to read honestly, please try to make them more clear or subgroup them
- the discussion does not seem like discussion, Authors should compare their results to others, I cannot find that
Author Response
Dear Reviewer,
Thank you for the encouragement and suggestions you have provided. I have incorporated them on the updated manuscript along with the suggestions from other reviewers:
- The title has been updated to “Effects of different trunk training methods for chronic low back pain: A Meta-analysis”.
- Reference for the “non-specific diagnosis” was the same as for “… can identify the cause of the pain 10% of the time”, hence the reference number was moved to the end of the sentence.
- Line 85-92 on methodology was added to provide further justification on the wide range of population age. And as reviewer mentioned, sub-group analysis of this study did show that younger CLBP patients are higher responders to training intervention compared to older CLBP patients. Therefore, in our discussion and results, we mentioned how future intervention and meta-analysis should clearly separate based on age group.
- we agree that the word "persistent" is debatable as in majority of CLBP cases, the pain is not constant, modulated by movement, posture, and rest. Constant pain itself according to American Family Physicians (Casazza, 2012) can be considered as red flag in low back pain screening. However, "persistent" is the word used by the European working group (reference 3, chapter 2) to define Chronic Non-Specific Low Back Pain which is why it was used in this article.
- We put borderline on figure 2-4, split and rearrange figure 4 into figure 4 and 5. We hope this improves the flow and readability.
- New line 335 to 339 that compared the result with results from 3 other studies.
We hope the included changes are sufficient to meet the journal standard.
Sincerely,
Authors
Reviewer 2 Report
There are some minor revisions that author(s) should made:
The authors of the research paper “Meta-analysis on the effects of different trunk training methods for chronic low back pain ” presented a topic relevant to low back pain; rehabilitation; exercise therapy, although many researchers have studied this topic and there are plenty of literature about it.
I do suggest to the authors that a review of very recent literature about the subject should be carried out. In fact, there are some recent and very relevant publications on this topic that should be included. Let me suggest some very recently references on this topic: Merinero, D.; Rodríguez-Aragón, M.; Álvarez-González, J.; López-Samanes, Á.; López-Pascual, J. Acute Effects of Global Postural Re-Education on Non-Specific Low Back Pain. Does Time-of-Day Play a Role? Int. J. Environ. Res. Public Health 2021, 18, 713. https://doi.org/10.3390/ijerph18020713; also Gonzalez-Medina, G.; Perez-Cabezas, V.; Ruiz-Molinero, C.; Chamorro-Moriana, G.; Jimenez-Rejano, J.J.; Galán-Mercant, A. Effectiveness of Global Postural Re-Education in Chronic Non-Specific Low Back Pain: Systematic Review and Meta-Analysis. J. Clin. Med. 2021, 10, 5327. https://doi.org/10.3390/jcm10225327, that could be included on your paper.
First of all, I would like to point out that materials, method and empirical results and statistics have been properly realized and the research methodology is appropriate.
However, the study focused on the working-age population (19–60 years) which is in fact, a very heterogeneous wide range of potential patients and some additional explanation about the subject is also required.
It is not clear enough for me the main goal of the paper and the introduction does not fix quite well with the main goal. I think the introduction should be rewritten to be more precise.
Finally, the conclusions are relevant but extremely short. Conclusions should be extended in order to provide with a more detailed final ideas, limitations and future researches should be included. Probably Conclusions should be rewritten including limitations and future researches on it.
Author Response
Dear Reviewer,
Thank you for the encouragement and suggestions you have provided. I have incorporated them on the updated manuscript along with the suggestions from other reviewers:
- The suggested articles are indeed related to the discussion, I have included them, along with another reference “Ikeda, D. M., & McGill, S. M. (2012). Can altering motions, postures, and loads provide immediate low back pain relief: a study of 4 cases investigating spine load, posture, and stability. Spine, 37(23), E1469-E1475.” On the discussion section, line 335 to 339.
- Line 85-92 on methodology was added to provide further justification on the wide range of population age. And as reviewer mentioned, sub-group analysis of this study did show that younger CLBP patients are higher responders to training intervention compared to older CLBP patients. Therefore, in our discussion and results, we mentioned how future intervention and meta-analysis should clearly separate based on age group.
- Last paragraph of introduction has been reworded to better match the discussion and conclusion.
- Conclusion was further elaborated based on your input.
We hope the included changes are sufficient to meet the journal standard.
Sincerely,
Authors
Reviewer 3 Report
The manuscript “Meta-analysis on the effects of different trunk training methods for chronic low back pain”, by Sutanto et al., reports a review and a comparative meta-analysis of different trunk training methods for chronic low back pain patients. Methods have been classified into three classes, and the effectiveness of interventions is compared between the different classes or between a class and a control group.
In my opinion, the authors are very clear in explaining the details of the methodology and of the selection of literature works, which are always an extremally delicate point in meta-analyses. Source databases and adopted guidelines are well described, as well as important details such as language of the work, geographic distribution, gender distribution. Exclusion criteria were also clearly stated, justifying to the strong selection realized on the large starting number of detected citations. The authors also described the different tests that can be employed to evaluate the effectiveness of a training method, underlining how sometime they can provide contradictory results. In addition, Table 1 in Appendix B (on the main features of the included trials) can represent, for a reader, a precious synthetic database of the up-today literature on the argument, and it is a signature of the huge work performed by the authors.
For all these reasons, I can recommend the publication of the manuscript on International Journal of Environmental Research and Public Health, provided that the following minor concerns are addressed.
- Can the author better specify what stated in lines 159-161, about the exclusion of the age-group of patients 40-45? It is not clear what they mean with “to remove subjectivity on study classification in the subgroup analysis”.
- In the same context, in line 161 it is written that two groups of patients, one under 40 and another one over 45, were excluded: why?
- In the comparative tables of figures 2-4, there is a column named “Weight”. I guess it is used to calculate a weighted average of MD. Can the authors specify how these weights are inferred? (maybe I missed it, but after looking inside the text I was unable to understand it)
- Can the authors better specify how they evaluated the risk of bias reported for each considered work, for different tasks, in figure 5? I saw some remarks in the main text; do they come from a subjective evaluation, work by work, by the authors? I suggest to include a short comment on the 21 “biased” works, maybe adding a column (“Detected bias risk”, for example) in Table 1 of Appendix B, or also creating a new small table.
Author Response
Dear Reviewer,
Thank you for the encouragement and suggestions you have provided. I have incorporated them on the updated manuscript along with the suggestions from other reviewers:
- Reworded line 164 – 169. The mean age 40 – 45 exclusion band from the subgroup analysis was to give a clear border line on which studies can be classified as using younger participants and which studies used older participants. There was ambiguity on how the participants in this age band should be classified, not to mention that most studies did not specify participant inclusion criteria based on age. With this exclusion band, we can confidently remove studies where participant mean age is 41±2 and mean age 44±2 from the subgroup analysis instead of subjectively arguing among authors on individual study classification in the subgroup analysis.
- Explanation of this question is on the above bullet point.
- Line 146-147 was added in the methods section provide added clarity on “weight” column on results
- Line 247-251 was added in the results section to provide added clarity on sensitivity analysis based on Risk of Bias 2.0 methodology. This added section was not written in the method section as a different reviewer commented that the method section was too long, some should be moved to results or discussion.
We hope the included changes are sufficient to meet the journal standard.
Sincerely,
Authors
Reviewer 4 Report
The paper by Sutanto and colleagues is well written and organized. The topic is interesting as CLBP affects people wordwide. The methodology is robust and valid. I only have two comments:
Methods:
- this section is too long. I suggest to reduce it and move some sentence in the discussion
- the authors should perform a GRADE of the included studies
Discussion:
this section should discuss the results and not only report them. I suggest to revise this section according to the previous comments.
Author Response
Dear Reviewer,
Thank you for the encouragement and suggestions you have provided. I have incorporated them on the updated manuscript along with the suggestions from other reviewers:
- Line 128-129 was abbreviated, line 172-174 was removed to reduce methods length. Other details had to be added to the methods section to provide clarification as indicated by other reviewers.
- GRADE analysis can provide more insight on the quality of evidence in meta-analysis. CLBP is an umbrella term of different pathologies such as spine disc herniation and zygapophyseal joint degeneration, where the condition may manifest differently in different age sub-groups. There are 3 intervention and control being compared in 3 different outcomes, with two subgroup analysis within this meta-analysis. The use of GRADE analysis may be limited for this study and hence we choose to use Cochrane Risk of Bias 2.0 only for each individual study analysis. Future meta-analysis with a more limited scope can benefit more from GRADE analysis.
We hope the included changes are sufficient to meet the journal standard.
Sincerely,
Authors
Round 2
Reviewer 4 Report
ok